# Bioactivity Features of a Zn-1%Mg-0.1%Dy Alloy Strengthened by Equal-Channel Angular Pressing

**DOI:** 10.3390/biomimetics8050408

**Published:** 2023-09-03

**Authors:** Natalia Martynenko, Natalia Anisimova, Maria Shinkareva, Olga Rybalchenko, Georgy Rybalchenko, Mark Zheleznyi, Elena Lukyanova, Diana Temralieva, Artem Gorbenko, Arseny Raab, Natalia Pashintseva, Gulalek Babayeva, Mikhail Kiselevskiy, Sergey Dobatkin

**Affiliations:** 1A.A. Baikov Institute of Metallurgy and Materials Science of the Russian Academy of Sciences, 119334 Moscow, Russia; n_anisimova@list.ru (N.A.); m.shinkareva29@mail.ru (M.S.); rybalch@mail.ru (O.R.); markiron@mail.ru (M.Z.); helenelukyanova@gmail.com (E.L.); diana4-64@mail.ru (D.T.); artemgorbenk@yandex.ru (A.G.); dobatkin.sergey@gmail.com (S.D.); 2N.N. Blokhin National Medical Research Center of Oncology (N.N. Blokhin NMRCO) of the Ministry of Health of the Russian Federation, 115478 Moscow, Russia; babaevagulyalek@gmail.com (G.B.); kisele@inbox.ru (M.K.); 3Center for Biomedical Engineering, National University of Science and Technology “MISIS”, 119049 Moscow, Russia; 4P.N. Lebedev Physical Institute of the Russian Academy of Sciences, 119991 Moscow, Russia; rybalchenkogv@lebedev.ru; 5Department of Physical Materials Science, National University of Science and Technology “MISIS”, 119049 Moscow, Russia; 6Institute of Innovative Engineering Technologies, Peoples’ Friendship University of Russia (RUDN University), 117198 Moscow, Russia; 7Institute of Physics of Advanced Materials, Ufa University of Science and Technology, 450076 Ufa, Russia; agraab@mail.ru; 8Limited liability Company “Veterinary Oncological Scientific Center”, 115211 Moscow, Russia; pashintseva2009@yandex.ru; 9Research Institute of Molecular and Cellular Medicine, Peoples’ Friendship University of Russia (RUDN University), 117198 Moscow, Russia

**Keywords:** zinc alloys, equal-channel angular pressing (ECAP), microstructure, mechanical properties, degradation, biocompatibility in vitro, biocompatibility in vivo

## Abstract

The structure, phase composition, corrosion and mechanical properties, as well as aspects of biocompatibility in vitro and in vivo, of a Zn-1%Mg-0.1%Dy alloy after equal-channel angular pressing (ECAP) were studied. The structure refinement after ECAP leads to the formation of elongated α-Zn grains with a width of ~10 µm and of Mg- and Dy-containing phases. In addition, X-ray diffraction analysis demonstrated that ECAP resulted in the formation of the basal texture in the alloy. These changes in the microstructure and texture lead to an increase in ultimate tensile strength up to 262 ± 7 MPa and ductility up to 5.7 ± 0.2%. ECAP slows down the degradation process, apparently due to the formation of a more homogeneous microstructure. It was found that the alloy degradation rate in vivo after subcutaneous implantation in mice is significantly lower than in vitro ones. ECAP does not impair biocompatibility in vitro and in vivo of the Zn-1%Mg-0.1%Dy alloy. No signs of suppuration, allergic reactions, the formation of visible seals or skin ulcerations were observed after implantation of the alloy. This may indicate the absence of an acute reaction of the animal body to the Zn-1%Mg-0.1%Dy alloy in both states.

## 1. Introduction

In recent years, zinc alloys have been considered promising materials for creating bioresorbable implants due to their acceptable degradation rate [1,2,3,4]. However, their level of biocompatibility still raises some concerns. For example, it was previously shown that the cytotoxicity of pure Zn with respect to MC3T3-E1 cells is unacceptably high [5]. However, the same study shows the excellent compatibility of zinc with HUVEC cells. On the other hand, the study [6] shows excellent hemocompatibility of pure Zn and an acceptable level of cytotoxicity with respect to MG63 cells. In addition, Zn demonstrated a high level of antibacterial activity against *Porphyromonas gingivalis* and an acceptable level of cytocompatibility [7]. The alloying of Zn with other biocompatible elements leads to a significant improvement of its biocompatibility. Yang et al. analyzed the biocompatibility of Zn-based binary alloys and showed that alloys of the Zn-Mg, Zn-Ca, Zn-Sr and Zn-Li systems have the highest level of cytocompatibility [5]. Zhang et al. [8] confirmed the conclusion about the good biocompatibility of Zn-Li alloys. They showed that the Zn-0.8%Li-0.2%Ag alloy does not adversely affect the proliferation of bone marrow mesenchymal stem cells. The alloys of the Zn-Sr system also demonstrated good hemocompatibility in vitro, a decrease in the viability of *S. aureus* in the culture medium and provided preconditions for bone formation and integration in vivo [9].

One of the most popular alloying systems for biomedical zinc alloys is the Zn-Mg system [10,11]. Magnesium is one of the most important elements necessary for the normal functioning of the body [12]. It is involved in various metabolic processes of the body, affecting the immune and nervous systems [12]. In addition, magnesium is an essential element for bone regeneration [13], and its consumption leads to an increase in bone mineral density [14]. Mg^2+^ ions with concentrations below 10 mM do not suppress the viability and osteogenic differentiation of human bone-marrow-derived stromal cells [15] and even positively affect the growth of mesenchymal stem cells [16]. Therefore, many studies are devoted to the study of the Zn-Mg system, including Zn-Mg-x. Previously, it was shown that Zn-Mg alloys can promote stem cell differentiation [17] and do not inhibit cell activity. In particular, Xiao et al. found no toxicity of pure Zn and the Zn-0.05Mg alloy with respect to mouse subcutaneous connective tissue L-929 line in vitro [18]. A number of studies, also conducted in vivo, showed satisfactory biocompatibility of alloys of the Zn-Mg system [19,20,21].

In this study, the aspects of biodegradation and biocompatibility in vitro and in vivo of the Zn-1 (wt.)%Mg-0.1 (wt.)%Dy alloy before and after equal-channel angular pressing (ECAP) were studied [22]. ECAP treatment makes it possible to improve the mechanical characteristics of zinc alloys due to structural and phase changes occurring during deformation [23]. At the same time, improved mechanical characteristics increase the prospects for using the studied alloy as a basis for the development of implants for osteosynthesis. The selection of the Zn-Mg-Dy alloy system is justified by the good biocompatibility observed in previous studies on Zn-Mg alloys [16,17,18,19,20,21]. The addition of Dy to the Zn-Mg alloy can also enhance its mechanical properties such as strength and hardness. This makes the Zn-Mg-Dy alloy system an attractive candidate for use in biomedical applications where both good biocompatibility and mechanical strength are required. Additionally, it is expected that the addition of dysprosium provides an acceptable level of corrosion resistance of the alloy with a good level of biocompatibility [24]. Previously, Dy^3+^ ions were shown to be strong inhibitors of the proliferation of B16 melanoma and L929 fibrosarcoma cells [25]. Feyerabend et al. also demonstrated the selective effect of Dy^3+^ ions on the viability of the MG63 human osteosarcoma cell line [26]. This specific effect may be of significant practical interest for use in clinical orthopedic oncology. However, further research is necessary for a complete understanding and optimization of the properties of the Zn-Mg-Dy alloy system; this should be aimed at studying the effect of deformation processes, including ECAP, on the operational characteristics of Zn-Mg-Dy alloys. It is expected that ECAP will significantly improve the mechanical properties of the studied alloy without compromising its corrosion resistance and biocompatibility. The improved mechanical properties of Zn-Mg-Dy alloys achieved through ECAP can greatly enhance the suitability of these alloys for various applications in clinical practice. This opens up new possibilities for their use for the creation of medical implants, orthopedic devices and other biomedical applications. For example, implants can be used for local immunotherapy of cancer patients, where an alloy scaffold can be used as a platform for drug delivery.

## 2. Materials and Methods

In this paper, the study of the effect of equal-channel angular pressing on the structure, mechanical properties, corrosion resistance and biocompatibility aspects of the Zn-1%Mg-0.1%Dy alloy was conducted. The melting of the alloy was carried out under laboratory conditions from pure zinc (99.975 wt.%), magnesium (99.95 wt.%) and dysprosium (99.9 wt.%). The melting of the alloy was conducted in an induction furnace without the use of a protective atmosphere. Then, the alloy was poured into a steel mold with a diameter of 35 mm and a height of 150 mm. Quantitative analysis of the alloy composition was performed on a BRUKER S8 Tiger serial X-ray fluorescence wave-dispersive spectrometer (series 2; Bruker, Karlsruhe, Germany). The actual content of alloying elements in the alloy was determined as 0.99 wt.%Mg and 0.11 wt.%Dy.

In the initial state, the alloy was homogenized at a temperature of 340 °C for 20 h and then water quenched. Further, the homogenized alloy was subjected to equal-channel angular pressing using the route Bc (channel intersection angle 120°). The deformation was carried out at a temperature of 200 °C. The number of ECAP passes (N) was 8, which corresponds to the degree of deformation equal to 7.

The alloy microstructure was studied using a JSM-7001F scanning electron microscope (SEM) (JEOL; Tokyo, Japan) equipped with an energy-dispersive spectrometer. The structure of the alloy after ECAP was studied in the longitudinal section. The size of structural elements was calculated using NEXSYS ImageExpert™ Pro 3 software (version 3; NEXSYS, Moscow, Russia). At least 20 pictures were used for this experiment. The number of structural elements measured to determine the average size of structural elements was at least 500. The X-ray diffraction (XRD) analysis was performed on a Bruker D8 Advance diffractometer (CuKα radiation, λ = 1.54 Å; Bruker, Karlsruhe, Germany). More details can be found in [27].

The mechanical properties of the alloy were determined by uniaxial tension on an Instron 3382 testing machine (Instron, High Wycombe, UK). The studies were carried out at room temperature with a deformation rate of 1 mm/min. The tests were carried out on flat specimens (3 samples per state) with a cross-sectional area of 2 mm × 1 mm and a working length of 5.75 mm.

Potentiodynamic research was made using an SP-300 potentiostat (Bio-Logic SAS, Seyssinet-Pariset, France). Studies were carried out in 0.9% NaCl solution (pH = 7) at room temperature. The setup used a flat PAR cell (Ametek Instruments, Oak Ridge, TN, USA) with a “three electrode configurations” (working electrode, Ag/AgCl reference electrode and Pt grid counter electrode). Scanning was carried out at a rate of 1 mV/s in the working range: OCP 150 mV, OCP + 500 mV, where OCP is the open circuit potential. The time to determine the open circuit potential was 10 min. Six scans were carried out for each state of the alloy. Corrosion potential and corrosion current density were calculated according to ASTM G59–97(2003) using EC-Lab software (BioLogic, Seyssinet-Pariset, France) [28].

The studies of the degradation rate and the biocompatibility of the alloy under in vitro and in vivo laboratory conditions were carried out on samples in the form of a parallelepiped with dimensions of 5 × 5 × 2 mm. For each study, three samples of each state of the alloy were used. Before research, the samples were sterilized by immersion in 70% ethanol for 2 h, followed by drying under sterile conditions.

The study of alloy degradation under in vitro conditions was carried out at a temperature of 37 °C in an atmosphere containing 5% carbon dioxide for 30 days. The tests were performed in a 24-well plate (Thermo Scientific, Roskilde, Denmark) in a complete growth medium based on Minimum Essential Medium Eagle (Sigma, Welwyn Garden City, UK) supplemented with 5% fetal bovine serum, 2 mM L-glutamine and 50 U/mL penicillin/streptomycin (PanEco, Moscow, Russia). The medium was renewed every 2 days. The samples before and after immersion tests were weighed on a Sartorius M2P Micro Balances Pro 11 electronic balance (Data Weighing Systems, Inc., Wood Dale, IL, USA; three digits per 1 mg). The degradation rate of the studied alloys was calculated according to the procedure described in [29].

The biocompatibility studies included an assessment of the hemolytic and cytostatic activity of the samples with respect to blood cells isolated from the peripheral blood of Balb/c mice using standard methodological approaches previously described in [30]. Red blood cells (RBC) were diluted with Hanks’s solution (PanEco, Moscow, Russia) to a concentration of 3 × 10^7^ cells per 1 mL in order to assess hemolysis. The 1.5 mL of RBC suspension was added to the samples. Then, the cell with the samples was incubated for 2 and 4 h at a temperature of 37 °C in an atmosphere of 5% carbon dioxide. At the end of the incubation, the cell suspension was centrifuged at 3000 rpm for 5 min, and then the optical density (OD) of the supernatant was measured in a 96-well plate (Nunc, Roskilde, Denmark) at 540 nm using a Spark plate reader (Tecan, Männedorf, Switzerland). The cell suspension incubated without samples served as a negative control (Control−). The cell suspension treated with Triton-X (Pan-Reac, AppliChem, Barcelona, Spain) was used as a positive control (Control+). The results of hemolysis study were calculated by the formula:$$Hemolysis\left(\%\right)=\frac{OD\left(Alloy\right)-OD(Control-)}{OD\left(Control+\right)-OD(Control-)}\times 100$$

To assess the cytotoxicity of samples, white blood cells (WBC) were suspended in a complete nutrient medium to a concentration of 6.72 × 10^5^ cells per 1 mL. The 1.5 mL of cell suspension was added to the samples. Then, the cell with the samples was incubated for 24 h under the same conditions as described above. After the end of the incubation period, cells were counted and their viability was assessed using the Muse Count & Viability Kit (Millipore, Burlington, MA, USA) using the Muse cell analyzer (Millipore, Darmstadt, Germany), following the manufacturer’s instructions. The WBC suspension incubated without samples was used as a control.

To evaluate the biocompatibility in vivo, samples of Zn-1%Mg-0.1%Dy alloys before and after ECAP were implanted under the skin of 10 Balb/c mice divided into two equal groups (males, m = 24 ± 2 g). The studies were carried out in accordance with the protocol approved by the decision of the Local Ethics Committee of “N.N. Blokhin National Medical Research Center of Oncology” of the Health Ministry of Russia (#05p-17/05/2023), based on the requirements of ISO 10993-2:2006 [31]. After 14 days, the animals were examined, assessing the integrity of the skin and the condition of the pelage. After euthanasia, samples of alloys were removed in order to study their rate of biodegradation. Also, blood serum samples of animals were collected to assess systemic toxicity and tissue adjacent to implants, in order to study cellular reactivity by studying morphology. Intacted Balb/c mice (*n* = 5, males, m = 23 ± 3 g) were used as control group.

The systemic toxicity of the alloy samples was assessed by comparing the concentration of bilirubin, urea, creatinine and albumin in the blood of mice with and without implants using an automatic biochemical analyzer EOS Bravo v.200 (Hospitex Diagnostics, Moscow, Russia).

The study of cellular reactivity in order to assess the biocompatibility of the samples was carried out by light microscopy of histological preparations of adipose tissue of mice from the area of implantation of the studied alloys. Tissue samples were fixed in 10% buffered formalin for 24 h, followed by posting the material on a Histo-Tek VP1 histological processor (Sakura Seiki Co., Ltd., Tokyo, Japan) and making paraffin blocks according to a standard protocol [32]. Automated hematoxylin-eosin (HE) staining of prepared tissue sections was carried out according to the classical method [33] on an automatic device for staining smears AFOMK-16-25-Pro (group of companies Emko, Moscow, Russia). The microscopy study was conducted using an Axioplan 2 device Imaging (Zeiss, Oberkochen, Germany).

Statistical analysis was performed based on the results of measurements in triplets by calculating the mean and standard deviation (m ± SD). A comparative analysis of the parameters characterizing the studied alloys was carried out using the *t*-test. Differences were considered significant at *p* < 0.05.

## 3. Results

Figure 1 shows the results of a study of the microstructure of the Zn-1%Mg-0.1%Dy alloy before and after ECAP. In the initial state, the alloy consists of α-Zn grains surrounded by an interlayer of phases. The average grain size of α-Zn is 30.7 ± 1.2 µm. The results of SEM-EDS mapping of the alloy sample showed a large amount of Mg in the grain boundary phase. In addition, it was possible to detect the presence of rather large Dy-containing particles (~10 μm). However, it should be noted that a small amount of Dy is also present in the α-Zn solid solution (Figure 1a,b).

ECAP is accompanied by both grain refinement and crushing of the grain boundary phase (Figure 1c,d). The general structure of the alloy after ECAP consisted of elongated grains with a size of ~10 µm and round inclusions of the Mg-containing phase with a size of 3–5 µm. The presence of Dy-rich particles was also confirmed by elemental SEM-EDS mapping.

X-ray diffraction (XRD) analysis was performed to determine the phase composition of the alloy (Figure 2 and Table 1). This showed that the alloy in both states consists of four phases: α-Zn, Mg_2_Zn_11_, MgZn_2_ and Dy_2_Zn_17_. At the same time, ECAP does not lead to a significant change in the volume fraction of phases (Table 1). However, it should be noted that a significant increase in the intensity of the (002)_α-Zn_ line with a decrease in the intensity of the (101)_α-Zn_ line is observed after ECAP. This may indicate the formation of a predominantly basal texture after ECAP.

Table 2 and Figure 3 present the measured mechanical characteristics of the alloy in the initial state and after ECAP. The results showed that the structural changes in the alloy after ECAP led to a simultaneous increase in both strength and ductility. Thus, the yield stress (YS) of the alloy after ECAP increased from 124 ± 18 to 212 ± 19 MPa, while the ultimate tensile strength (UTS) increased from 132 ± 18 to 262 ± 7 MPa. It is interesting that the deformation also leads to a significant increase in the ductility of the alloy, raising the elongation (El) from 0.8 ± 0.5% to 5.7 ± 0.2%.

Figure 4 shows the results of a study of the corrosion resistance of the Zn-1%Mg-0.1%Dy alloy before and after ECAP.

The results of the study of electrochemical corrosion showed that ECAP does not affect the alloy’s resistance to electrochemical corrosion. The corrosion potential does not differ significantly and amounts to −1001 ± 34 and −1010 ± 15 mV for the alloy in the initial and ECAP-treated states, respectively. At the same time, structural changes caused by ECAP lead to a significant decrease in the electrochemical corrosion rate, as indicated by a decrease in the corrosion current density. The corrosion current density of the alloy is 13.7 ± 3.4 and 1.6 ± 0.6 μA/cm^2^ for the initial and deformed states, respectively (Figure 4a,b). At the same time, the study of the degradation rate, both under in vitro and in vivo conditions, shows a good agreement with the results of the study of electrochemical corrosion (Figure 4c). ECAP leads to a decrease in the degradation rate of alloy in both cases. However, the degradation process is slowed down during in vivo degradation. The average degradation rate of the alloy after 30 days of incubation in a complete growth medium based on Minimum Essential Medium Eagle is 0.30 ± 0.03 and 0.19 ± 0.05 mm/year for the initial and ECAP-treated states, respectively. The average degradation rate in vivo after 14 days of implantation is 0.14 ± 0.02 and 0.08 ± 0.01 mm/year for the alloy before and after ECAP, respectively.

Hemolytic activity and cytotoxicity studies were carried out to evaluate the biocompatibility in vitro of the Zn-1%Mg-0.1%Dy alloy in the initial state and after ECAP. It was found that hemolysis does not exceed 5%. This means that the samples are biocompatible with respect to RBC (Figure 5) according to the recommendations of ISO 10993-5, ISO 10993-4 and ASTM F 756-00 [34,35,36,37].

Cytotoxicity of samples was investigated through flow cytometry by counting cells, as well as by assessing cell viability after incubation with alloy samples for 24 h (Figure 6). The obtained results demonstrate the absence of signs of cytotoxicity of the alloy in both states, since the comparative analysis did not show significant differences from the control (*p* > 0.05). The same conclusions were made comparing groups of samples before and after ECAP (*p* > 0.05). The above results confirm that the alloys in both states do not cause destruction of blood cells, which proves their biocompatibility in vitro.

The biocompatibility in vivo was studied by implanting samples of the alloy under the skin of mice. The reactivity of subcutaneous tissue cells to contact with the studied alloy was assessed using histological methods after 14 days. In addition, blood biochemical parameters reflecting changes in the function of the animal’s liver and kidneys were studied. These could be affected by the process of the excretion of alloy biodegradation products from the body or the development of immunological reactions.

An examination of the animals of both groups did not demonstrate signs of suppuration, allergic reactions, the formation of visible seals or skin ulcerations after 2 weeks of implantation of the alloys. Indirectly, this indicates the absence of an acute reaction of the body to the Zn-1%Mg-0.1%Dy alloy before and after ECAP. The results of the study of tissue morphology demonstrate the formation of a connective tissue capsule around the alloy samples at the border of their contact with subcutaneous tissue (Figure 7). The capsule was formed by areas of reactive collagenous tissue, penetrated by small-caliber blood vessels, with focal infiltration by macrophages and a smaller number of small lymphocytes in the area of alloy sample/tissue contact. The formation of an abscess with infiltration by neutrophils was not observed. The presence of mononuclear cells usually indicates the development of the body’s reaction to the implantation of a foreign material and the development of its biodegradation process due to cellular biodestruction. The activation of the biodegradation process in the area of implantation is also evidenced by the accumulation of biodegradation products and the presence of dilated blood vessels, both in the subcutaneous tissue surrounding the alloy and in the capsule around it. Since a general examination of the animals’ state did not reveal signs of acute rejection or inflammation, the described changes in the cellular composition were considered a standard reaction, associated with the formation of granulation tissue in the area of the surgical operation, with a tendency to normal healing of the damaged tissue and gradual cleansing of foreign material from the tissues. Comparing the severity of the described features in both groups shows that the degradation of the Zn-1%Mg-0.1%Dy alloy after ECAP proceeded less intensively. This is confirmed by a more significant accumulation in tissues of the alloy degradation products in the initial state in the form of crystals. This in turn stimulates an enhanced cellular response involving a mass of macrophages in the area of implantation and the formation of a capsule with thickened walls penetrated by dilated blood vessels. It is important to note that such vessels are characteristic of granulation tissue.

The systemic toxic effect was characterized by assessing the functioning of the main organs of excretion, the kidneys and liver. The influence of the formation of a large number of blood vessels in the area of implantation, which contribute to the elimination of biodegradation products of the alloy of implanted samples, was taken into account. For this study, a number of biochemical parameters of blood serum were evaluated. To study liver function, bilirubin and albumin in blood serum were evaluated, and urea and creatinine were evaluated for kidney function assay. The obtained data were compared with the results of the control group (Table 3). This result demonstrates the absence of significant differences in biochemical parameters in animals after the implantation of alloy samples compared with the control group (*p* > 0.05). It proves that the biodegradation of Zn-1%Mg-0.1%Dy alloy samples in the initial state and after ECAP does not have a systemic toxic effect on the functioning of the main organs of the animal body and does not induce the development of acute reactive conditions by the foreign body rejection mechanism. This allows us to state that the ECAP-treated Zn-1%Mg-0.1%Dy alloy is biocompatible in vivo.

## 4. Discussion

In this work, the effect of ECAP on the mechanical properties, degradation rate and biocompatibility aspects of the Zn-1%Mg-0.1%Dy alloy was studied. It was shown that ECAP leads to a significant increase in the strength (YS up to 212 ± 19 MPa, UTS up to 262 ± 7 MPa) and ductility of the alloy (up to 5.7 ± 0.2%). The increase in these parameters is apparently associated with structural and textural changes after deformation. The grain refinement by three times led to a twofold increase in strength. In addition, the refinement and rearrangement of the grain boundary phases also have a positive effect. In the initial state, a continuous network of Mg-containing phases is located along the α-Zn boundaries. After ECAP, the grain boundary phase is crushed and converted into separate, mostly rounded particles. The transformation of the grain boundary phases has a positive effect on the ductility of the alloy. An additional reason for the increase in ductility is probably the textural changes that occurred after ECAP. Zhuo et al. previously showed that ECAP can increase the strength and ductility of the Zn-2.5Ag-0.08Mg alloy up to 393.1 MPa and 56.3%, respectively [38]. This combination of strength and ductility is due to strong grain refinement, suppression of deformation twinning and activation of <c + a> pyramidal sliding. In the Zn-0.5Ag-0.08Mg alloy, ECAP leads to an increase in strength and ductility up to 388.1 MPa and 45.3% due to grain refinement, dislocation hardening, the formation of a relatively weak texture and the suppression of twinning [23]. Liu et al. also showed that 12 passes of ECAP at 150 °C can increase the strength of the Zn-1.6Mg alloy up to 423 MPa at an elongation level of 5.2% [39].

An improvement in the corrosion resistance of the alloy is an additional positive effect of ECAP. Thus, the degradation rate of alloy after ECAP decreased from 0.30 ± 0.03 to 0.19 ± 0.05 mm/year after 30 days of incubation in vitro in the complete growth medium and from 0.14 ± 0.02 to 0.08 ± 0.01 mm/year after 14 days of alloy implantation. Taking into account the lack of change in the phase composition of the alloy and the volume fraction of these phases after ECAP, the deceleration of the degradation process can be explained by the formation of a uniform microstructure and fragmentation of the grain boundary phase. The difference in the degradation rate measured under in vitro and in vivo conditions may be due to different levels of aggressiveness of the medium. In the case of immersion tests, there is a gradual accumulation of degradation products and metal ions in the test medium, while no such pattern is observed during implantation. The additional influence of the atmosphere on the degradation process can be excluded under in vivo conditions. Nevertheless, body fluids composition, which can have a negative effect on corrosion rate, should be taken into account for in vivo degradation. Therefore, the difference (most often a slowdown) in the rates of degradation in vitro and in vivo is a typical situation for biodegradable materials [40]. For example, Sun et al. found that the degradation of the Zn–Fe alloy under in vitro conditions is significantly lower than the degradation rate upon implantation into the subcutaneous and femoral tissues [41]. At the same time, Shao et al. observed a gradual slowdown in the degradation rate during implantation in the frontal bone, mandible and femur in beagles (from 0.183 mm/year in the first 3 months to 0.065 mm/year after 12 months) [19].

The Zn-1%Mg-0.1%Dy alloy was considered promising for use in medicine as the basis for implantable structures for orthopedics. Therefore, the change in its biocompatibility in vitro and in vivo in comparison with the initial state was evaluated. The studies of hemolysis and cytotoxicity in vitro showed that ECAP of the Zn-1%Mg-0.1%Dy alloy did not impair its properties in vitro, i.e., did not destroy cells. However, the level of hemolysis induced by the ECAP-treated alloy after 4 h of incubation exceeds the effect of the alloy in the initial state, although it did not exceed 5%. Given the fact that the degradation rate of the initial alloy is higher than that of the ECAP-treated alloy, the exact reason for such behavior remains unclear. It is possible that the accumulation of a higher concentration of Zn^2+^ ions in the growth medium due to the faster degradation process could have a positive effect on RBC. Earlier studies have shown that low concentrations of Mg^2+^ ions can have a positive effect on the growth of mesenchymal stem cells [16]. However, a high concentration of magnesium ions can enhance hemolytic activity [42]. In addition, zinc ions increase the osmotic stability of the RBC membrane [43]. This has also been demonstrated for cell adhesion [5]. However, increasing the concentration of Zn^2+^ ions beyond the permissible limit reduces these indicators. Therefore, a negative effect of ECAP-treated alloy on cells is more likely after contact with the alloy in the initial state, where degradation is significantly faster. It can be expected that the high degradation rate may lead to a decrease in biocompatibility indicators with an increase in incubation time.

To assess biocompatibility in vivo, alloy samples were implanted under the skin of laboratory animals, which subsequently did not cause an acute reaction of the body to the Zn-1%Mg-0.1%Dy alloy either before or after ECAP. This result is in good agreement with the one obtained by Drelich et al., who showed that Zn wire implanted in a mouse artery exhibited stable corrosion without local toxicity for at least 20 months after implantation [44]. The study of the development of the local reaction of the animal body by histological methods revealed the encapsulation of samples with the involvement of macrophages in the area of implantation, which is considered a standardized reaction to a foreign body [45]. In this case, the reaction to the alloy after ECAP was less pronounced in comparison with the reactivity with respect to the initial-state alloy. It is obvious that the reason for these differences could be the higher rate of biodegradation of the alloy in the initial state. The higher degradation rate was accompanied by the accumulation of a larger amount of degradation products that formed crystals inside the capsule and the attraction of a large number of effector cells into the volume and walls of the capsule. The formation of similar needles was observed by Shao et al. [19]. To assess the systemic toxicity of biodegradation products on the body, the biochemical parameters of the blood, reflecting changes in the functions of main excretory organs such as the liver and kidneys, were studied. The performed analysis showed no significant changes in bilirubin, albumin, urea and creatinine after implantation of samples of Zn-1%Mg-0.1%Dy alloy before and after ECAP in comparison with intact mice of the control group. On this basis, it can be concluded that there was no pronounced systemic acute and subacute toxic effect of these biodegradable samples on experimental animals during the observation process. In general, the results of the biological properties study of the alloy samples in vitro and in vivo allow us to conclude that ECAP of the Zn-1%Mg-0.1%Dy alloy does not impair the biocompatibility of the samples, while improving the mechanical characteristics.

As is known, biocompatibility is an indispensable requirement for medical materials in general, and implantable structures for osteosynthesis in particular. In case of metal constructs for osteosynthesis development, along with biocompatibility, the material for these products should be characterized by a prolonged biodegradation rate in order to ensure stable synthesis until the completion of bone defect repair. Taking this into account, the obtained data allow us to consider the Zn-1%Mg-0.1%Dy alloy after ECAP as a promising basis for creating metal devices for osteosynthesis. The next step of this research should be the study of the kinetics of biodegradation of samples for orthopedics based on the Zn-1%Mg-0.1%Dy alloy after ECAP. It should be verified that there is no chronic systemic toxicity during a long period of implantation of the developed metal constructs in vivo, which is necessary for the successful completion of the process of osteosynthesis and reconstruction of the bone defect.

## 5. Conclusions

The effect of ECAP on the mechanical properties, corrosion resistance and biocompatibility aspects of a potential medical Zn-1%Mg-0.1%Dy alloy were studied. The choice of alloying elements was due to their positive effect on mechanical and corrosion properties. In addition, it was expected that Dy^3+^ ions released during degradation could inhibit the viability of tumor cells. The main conclusions that can be drawn from this study are as follows:ECAP leads to grain refinement of the Zn-1%Mg-0.1%Dy alloy and crushing of the magnesium phases without changing the phase composition of the alloy.A significant increase in strength and ductility is observed in the Zn-1%Mg-0.1%Dy alloy after ECAP. The ultimate tensile strength of the alloy after ECAP was 262 ± 7 MPa, while the ductility was 5.7 ± 0.2%.ECAP slows down the rate of electrochemical corrosion of the alloy, which facilitates the slowdown of its biodegradation after implantation in comparison with the alloy in the initial state. The degradation rate of the alloy after ECAP decreases both for in vitro and in vivo conditions. At the same time, the degradation rate under in vivo conditions is significantly reduced compared to the values measured in vitro.ECAP does not impair the in vitro and in vivo biocompatibility of the Zn-1%Mg-0.1%Dy alloy.

Thus, our study showed that the use of ECAP can significantly improve the mechanical properties, including ductility, of medical Zn-based alloys without compromising their corrosion resistance and biocompatibility in vitro and in vivo. ECAP makes it possible to double the strength of the Zn-1%Mg-0.1%Dy alloy with an increase in ductility by approximately seven times due to the grain refinement and texture transformation. The obtained data indicate the prospects for the development of biodegradable implants fabricated from ECAP-treated Zn-based alloys. In particular, the obtained results justify the use of ECAP-treated Zn alloys as the basis for submersible implants and fasteners for osteoreconstructive surgery.

## Figures and Tables

**Figure 1 biomimetics-08-00408-f001:**
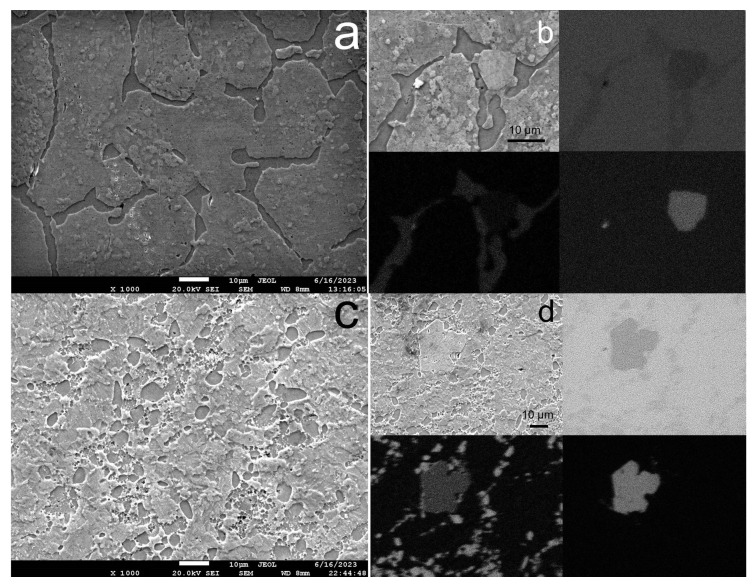
SEM-image (SE-contrast) of microstructure of the Zn-1%Mg-0.1%Dy alloy before (**a**,**b**) and after ECAP (**c**,**d**) with elemental SEM-EDS mapping (**b**,**d**).

**Figure 2 biomimetics-08-00408-f002:**
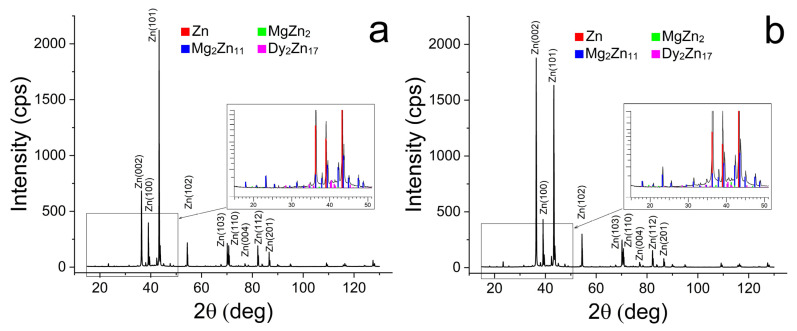
X-ray diffraction patterns of the Zn-1%Mg0.1%Dy alloy in initial (**a**) and ECAP-treated (**b**) states.

**Figure 3 biomimetics-08-00408-f003:**
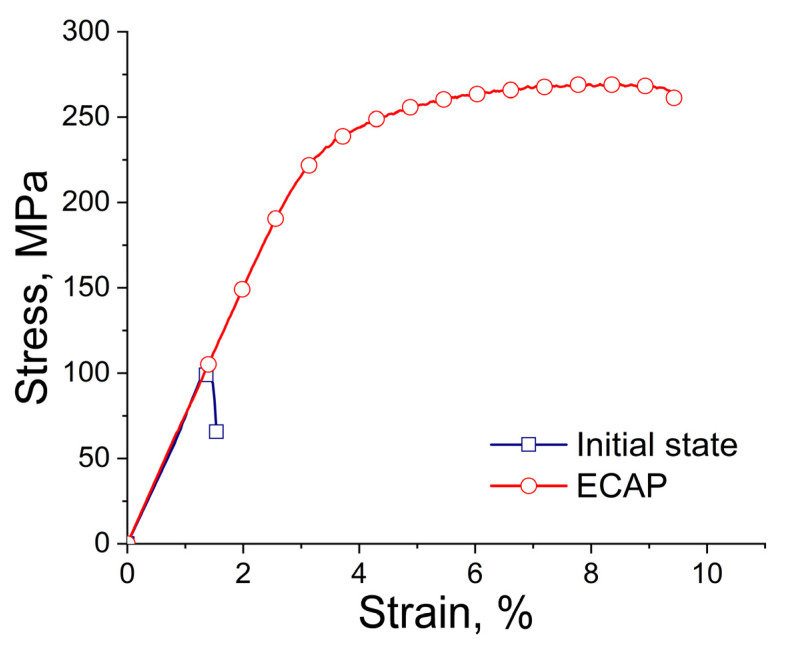
Stress–strain curves for the Zn-1%Mg-0.1%Dy alloy before and after ECAP.

**Figure 4 biomimetics-08-00408-f004:**
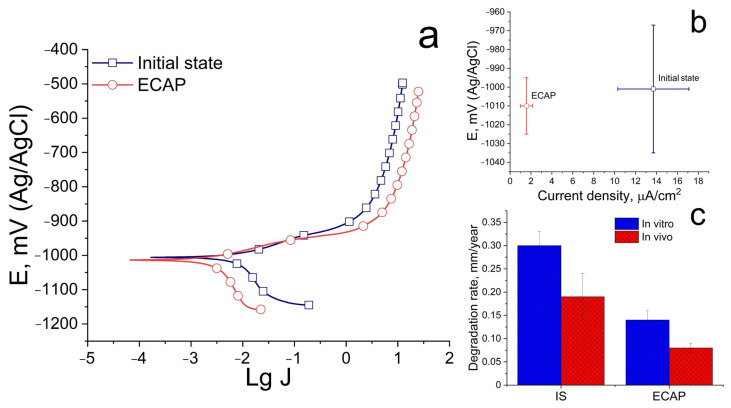
Polarization curves (Ag/AgCl electrode; J—corrosion current density) in 0.9% NaCl solution (**a**), the values of the corrosion potential and the corrosion current density (**b**) and in vitro and in vivo degradation rate (**c**) of the Zn-1%Mg-0.1%Dy alloy before and after ECAP (IS—initial state).

**Figure 5 biomimetics-08-00408-f005:**
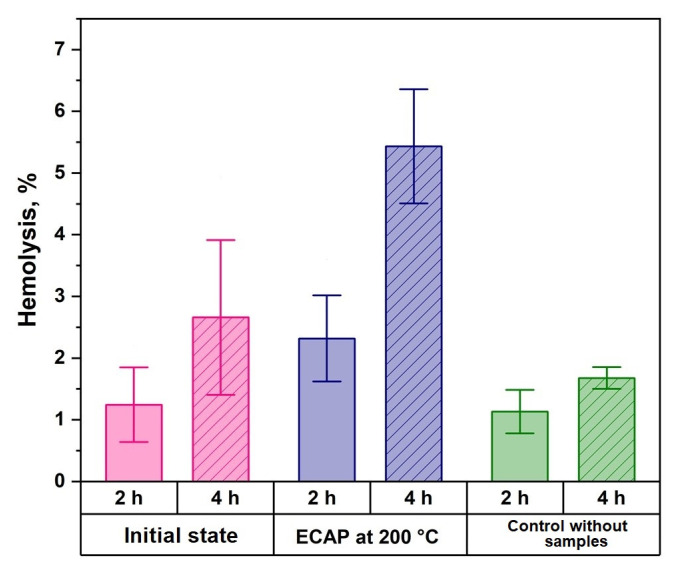
The results of the study of hemolysis of Zn-1%Mg-0.1%Dy alloys in initial and ECAP-treated states in comparison with the control.

**Figure 6 biomimetics-08-00408-f006:**
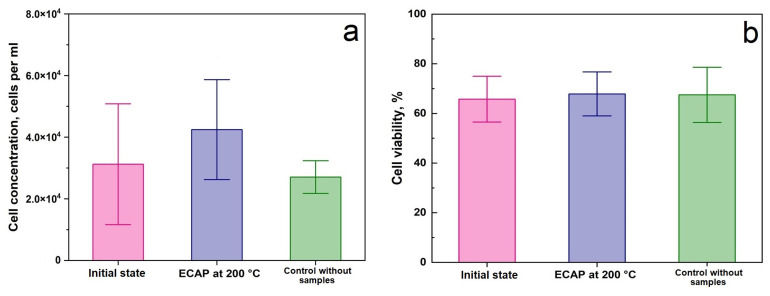
The results of the study of the cytotoxicity of Zn-1%Mg-0.1%Dy alloy in initial state and after ECAP: cell concentration (**a**) and cell viability (**b**) in comparison with control (without samples).

**Figure 7 biomimetics-08-00408-f007:**
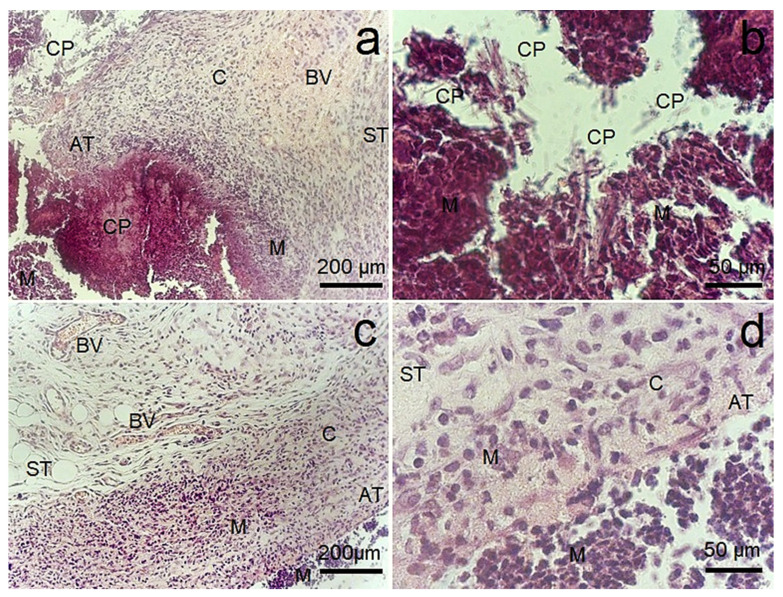
Mouse subcutaneous fat (ST) adjacent to Zn-1%Mg-0.1%Dy implants in the initial state (**a**,**b**) and after ECAP (**c**,**d**); HE staining. Magnification ×200 (**a**,**c**), ×1000 (**b**,**d**). The formation of blood vessels (BV) and accumulation of macrophages (M) were observed in a capsule (C) formed at the alloy sample/tissue contact (AT). An accumulation of alloy corrosion products (CP) surrounded by macrophages is observed near the inner side of the capsule.

**Table 1 biomimetics-08-00408-t001:** Results of XRD analysis (Rietveld method) of the Zn-1%Mg-0.1%Dy alloy before and ECAP.

State	Phase	Space Group	Wt.% Rietveld	Cell Volume (Å^3^)	a (Å)	c (Å)
Initial state	Zn	P6_3_/mmc (194)	93.5 ± 0.9	30.423 ± 0.001	2.665 ± 0.001	4.948 ± 0.001
	Mg_2_Zn_11_	Pm$\bar{3}$ (200)	3.5 ± 0.5	622.919 ± 0.088	8.540 ± 0.005	8.540 ± 0.005
	MgZn_2_	P6_3_/mmc (194)	1.0 ± 0.6	182.493 ± 0.174	4.897 ± 0.001	8.788 ± 0.008
	Dy_2_Zn_17_	R$\bar{3}$m (166)	2.0 ± 0.8	892.169 ± 0.980	8.890 ± 0.007	13.036 ± 0.020
ECAP	Zn	P6_3_/mmc (194)	93.0 ± 1.1	30.418 ± 0.001	2.665 ± 0.001	4.947 ± 0.001
	Mg_2_Zn_11_	Pm$\bar{3}$ (200)	3.5 ± 0.9	622.719 ± 0.083	8.539 ± 0.005	8.539 ± 0.005
	MgZn_2_	P6_3_/mmc (194)	1.1 ± 0.8	181.891 ± 0.751	4.896 ± 0.009	8.764 ± 0.017
	Dy_2_Zn_17_	R$\bar{3}$m (166)	2.4 ± 1.1	898.121 ± 1.713	8.921 ± 0.015	13.031 ± 0.052

**Table 2 biomimetics-08-00408-t002:** Mechanical properties of the Zn-1%Mg-0.1%Dy alloy before and after ECAP.

Processing	YS, MPa	UTS, MPa	El, %
Initial state	124 ± 18	132 ± 18	0.8 ± 0.5
ECAP	212 ± 19	262 ± 7	5.7 ± 0.2

**Table 3 biomimetics-08-00408-t003:** Blood biochemical parameters of mice after implantation of Zn-1%Mg-0.1%Dy alloys in initial and ECAP-treated states in comparison with intact animals (2 weeks after sample implantation).

Parameter	Intact Animals	Initial State	*p* *	ECAP	*p* *
Bilirubin, μmol/L	5.6 ± 3.5	4.8 ± 3.8	0.59	5.8 ± 0.2	0.88
Urea, U/L	4.4 ± 4.1	7.7 ± 0.1	0.52	8.1 ± 3.5	0.54
Creatinine, U/L	78.7 ± 10.5	88.3 ± 2.2	0.51	78.0 ± 2.7	0.91
Albumin, g/L	38.9 ± 8.1	37.8 ± 6.4	0.77	36.0 ± 4.2	0.67

* Zn-1%Mg-0.1%Dy alloy in initial state and after ECAP vs. intact animals.

## Data Availability

All the data required to reproduce these experiments are present in the article.

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
