# Peer review of "Bioactivity Features of a Zn-1%Mg-0.1%Dy Alloy Strengthened by Equal-Channel Angular Pressing"

_biomimetics, 2023, doi:10.3390/biomimetics8050408_

Round 1
Reviewer 1 Report
I support publishing the manuscript after moderate editing of the English language.
Moderate editing of the English language is required.
Author Response
Reviewer #1: I support publishing the manuscript after moderate editing of the English language.
Answer: We thank the reviewer for appreciating our article. An extensive editing of English language has also been made.

Reviewer 2 Report
Authors of Biomimetics manuscript 2555764: (1) nice work; well written; excellent Materials/Methods section. (2) In the Materials and Methods section, add the number of tension specimens that were tested for the two cases (Initial state and ECAP) - I assume you took an average of several tests to get the YS, UTS and %El numbers in Table 2. You should cover this same issue for the other measurements you took - number of tests to arrive at presented (average) values? (3) Big problem. Your Conclusions section does not include any "conclusions" which are defined as generalizations based on the results. You have listed only results. To increase the impact of this work you should develop one or more conclusions and mention the most important one in the abstract. For instance, if grain refinement and improved microstructural homogeneity caused by ECAP (or some other severe plastic deformation (SPD) process) improve the biocompatibility of your Zn-1%Mg-0.1%Dy alloy, will not other implant alloys also see improved biocompatibility following SPD treatment? This could be a conclusion (yet to be proven, but a sound hypothesis). Such a statement will bring increased attention to the work. Think about it.
Author Response
Reviewer #2: Authors of Biomimetics manuscript 2555764: (1) nice work; well written; excellent Materials/Methods section.
(2) In the Materials and Methods section, add the number of tension specimens that were tested for the two cases (Initial state and ECAP) - I assume you took an average of several tests to get the YS, UTS and %El numbers in Table 2. You should cover this same issue for the other measurements you took - number of tests to arrive at presented (average) values?
Answer: The corrections have been made in «Materials and methods» section:
«At least 20 pictures were used to calculate the average value of structural elements. The number of structural elements measured to calculate the average value was at least 500».
«The mechanical properties of the alloy were determined by uniaxial tension on an Instron 3382 testing machine (Instron, High Wycombe, UK). The studies were carried out at room temperature with a deformation rate of 1 mm/min. The tests were carried out on flat specimens (3 samples per state) with a cross-sectional area of 2 mm × 1 mm and a working length of 5.75 mm».
It was written for PDP and degradation and biocompatibility tests:
«Six scans were carried out for each state of the alloy» - PDP tests
«The studies of the degradation rate and the biocompatibility of the alloy under in vitro and in vivo conditions were carried out on samples in the form of a parallelepiped with dimensions of 5 × 5 × 2 mm» - Degradation and biocompatibility tests
(3) Big problem. Your Conclusions section does not include any "conclusions" which are defined as generalizations based on the results. You have listed only results. To increase the impact of this work you should develop one or more conclusions and mention the most important one in the abstract. For instance, if grain refinement and improved microstructural homogeneity caused by ECAP (or some other severe plastic deformation (SPD) process) improve the biocompatibility of your Zn-1%Mg-0.1%Dy alloy, will not other implant alloys also see improved biocompatibility following SPD treatment? This could be a conclusion (yet to be proven, but a sound hypothesis). Such a statement will bring increased attention to the work. Think about it.
Answer: Conclusions was revised according the comment:
«The effect of ECAP on the mechanical properties, corrosion resistance and aspects of biocompatibility of a potential medical Zn-1%Mg-0.1%Dy alloy were studied. The choice of alloying elements was due to their positive effect on mechanical and corrosion properties. In addition, it was expected that Dy3+ ions released during degradation could inhibit the viability of tumor cells. The main conclusions that can be drawn from this study are as follows:
- ECAP leads to grain refinement of the Zn-1%Mg-0.1%Dy alloy and crushing of the magnesium phases without changing the phase composition of the alloy.
- A significant increase in strength and ductility is observed in the Zn-1%Mg-0.1%Dy alloy after ECAP. The ultimate tensile strength of the alloy after ECAP was 262 ± 7 MPa, while the ductility was 5.7 ± 0.2%.
- ECAP slows down the rate of electrochemical corrosion of the alloy, which mediates the slowdown of its biodegradation after implantation in comparison with the alloy in the initial state. The degradation rate of the alloy after ECAP decreases both for in vitro and in vivo conditions. At the same time, the degradation rate under in vivo conditions is significantly reduced compared to the values measured in vitro.
- ECAP does not impair biocompatibility in vitro and in vivo of the Zn-1%Mg-0.1%Dy alloy.
Thus, our study showed that the use of ECAP can significantly improve the mechanical properties, including ductility, of medical Zn-based alloys without compromising their corrosion resistance and biocompatibility in vitro and in vivo. ECAP makes it possible to double the strength of the Zn-1%Mg-0.1%Dy alloy with an increase in ductility by approximately 7 times due to the grain refinement and texture transformation. The obtained data indicate the prospects for the development of biodegradable implants fabricated from the ECAP-treated Zn-based alloys. In particular, the obtained results justify the use of ECAP-treated Zn alloys as the basis for submersible implants and fasteners for osteoreconstructive surgery».

Reviewer 3 Report
This study is well conducted organizing various investigations of mechanical testing, electro-chemical measurement, In-vitro and In-vivo examinations. However, ECAP is a well known technique to improve mechanical strength of light metals by controlling crystal grain size, and grain boundary. Therefore, the biodegradation and biocompatibility of the Zn-1%Mg-0.1%Dy should be highlighted in this article. Please consider the following review comments to enhance the value of this article.
The introduction is lengthly, the early part obstructed to understand the intention of this study. The author should focus on the clinical significance of Zn-1%Mg-0.1%Dy alloy, and clearly describe the issues to be achieved, and the expected application of the proposed ECAP alloy, especially onco-orthopedics.
Line 362 misspelling ‘in virot’, should be ‘in vitro’.
The words ‘Annealing’ and ‘Initial state’ are confusedly used in the figures and the captions, which should be consistent.
In the hemolytic activity and cytotoxicity studies, the word ‘the control’ is uncertain, which should be kindly described as ‘the control without samples’ in the figure captions and the main text.
The author should discuss the reason for the increase of hemolysis in ECAP alloy, which is demonstrated in Figure 5.
Regarding the in vivo study, the site of under the skin is a wet environment but supposingly poor blood flow as compared with bone and intramedullary parts. Because the degradation/corrosion rate of biodegradable alloy is much influenced by the implant site, the author should discuss this point of view supposing specific application of the proposed materials.
Please simplify the sentence and paragraph to help clear understanding.
Author Response
Reviewer #3: This study is well conducted organizing various investigations of mechanical testing, electro-chemical measurement, In-vitro and In-vivo examinations. However, ECAP is a well-known technique to improve mechanical strength of light metals by controlling crystal grain size, and grain boundary. Therefore, the biodegradation and biocompatibility of the Zn-1%Mg-0.1%Dy should be highlighted in this article. Please consider the following review comments to enhance the value of this article.
The introduction is lengthly, the early part obstructed to understand the intention of this study. The author should focus on the clinical significance of Zn-1%Mg-0.1%Dy alloy, and clearly describe the issues to be achieved, and the expected application of the proposed ECAP alloy, especially onco-orthopedics.
Answer: Introduction was revised according the comment:
«In this work, the aspects of biodegradation and biocompatibility in vitro and in vivo of the Zn-1 (wt.)%Mg-0.1 (wt.)%Dy alloy before and after equal-channel angular pressing (ECAP) were studied [22]. ECAP treatment makes it possible to improve the mechanical characteristics of zinc alloys due to structural and phase changes occurring during deformation [23]. At the same time, improved mechanical characteristics increase the prospects for using the studied alloy as a basis for the development of implants for osteosynthesis. The selection of the Zn-Mg-Dy alloy system is justified by the good biocompatibility observed in previous research studies on Zn-Mg alloys [16-21]. The addition of Dy to the Zn-Mg alloy can also enhance its mechanical properties such as strength and hardness. This makes the Zn-Mg-Dy alloy system an attractive candidate for use in biomedical applications where both good biocompatibility and mechanical strength are required. Besides, it is expected that the addition of dysprosium provide an acceptable level of corrosion resistance of the alloy with a good level of biocompatibility [24]. Previously Dy3+ ions were shown to be strong inhibitors of the proliferation of B16 melanoma and L929 fibrosarcoma cells [25]. Feyerabend et al. also demonstrated the selective effect of Dy3+ ions on the viability of the MG63 human osteosarcoma cell line [26]. This specific effect may be of significant practical interest for use in clinical orthopedic oncology. However, further research is necessary for a complete understanding and optimization of the properties of the Zn-Mg-Dy alloy system, aimed at studying the effect of deformation processes, including ECAP, on operational characteristics of Zn-Mg-Dy alloys. It is expected that ECAP will significantly improve the mechanical properties of the studied alloy without compromising its corrosion resistance and biocompatibility. The improved mechanical properties of Zn-Mg-Dy alloys achieved through ECAP can greatly enhance the suitability of these alloys for various applications in clinical practice. This opens up new possibilities for their use for creation of medical implants, orthopedic devices, and other biomedical applications. As an example implants for local immunotherapy of cancer patients where alloy scaffold can be used as platform for drug delivery».
Line 362 misspelling ‘in virot’, should be ‘in vitro’.
Answer: The typo was corrected.
The words ‘Annealing’ and ‘Initial state’ are confusedly used in the figures and the captions, which should be consistent.
Answer: The alloy before ECAP was named as the «initial state» in the text of the article
In the hemolytic activity and cytotoxicity studies, the word ‘the control’ is uncertain, which should be kindly described as ‘the control without samples’ in the figure captions and the main text.
Answer: The figures 5 and 6 were corrected according the comment:
The author should discuss the reason for the increase of hemolysis in ECAP alloy, which is demonstrated in Figure 5.
Answer: The probable reason of increase in hemolysis level of the alloy after ECAP was analyzed in the "Discussion" section. « The studies of hemolysis and cytotoxicity in vitro showed that ECAP of the Zn-1%Mg-0.1%Dy alloy did not impair its properties in vitro, i.e., did not destroy cells. However, the level of hemolysis induced by ECAP-treated alloy after 4 hours of incubation exceeds the effect of the alloy in initial state although it did not exceed 5%. Given the fact that the degradation rate of the initial alloy is higher than that of the ECAP-treated alloy, the exact reason for such behavior remains unclear. It is possible that the accumulation of a higher concentration of Zn2+ ions in the growth medium due to the faster degradation process could have a positive effect on RBC. Earlier studies have shown that low concentrations of Mg2+ ions can have a positive effect on the growth of mesenchymal stem cells [Maradze, D.; Musson, D.; Zheng, Y.; Cornish, J.; Lewis, M.; Liu Y. High Magnesium Corrosion Rate has an Effect on Os-teoclast and Mesenchymal Stem Cell Role During Bone Remodelling. Sci. Rep. 2018, 8, 10003. https://doi.org/10.1038/s41598-018-28476-w]. However, a high concentration of magnesium ions can enhance hemolytic activity [Fife, E.H.Jr.; Muschel, L.H. Influence of Magnesium and Calcium Ions on the Hemolytic Activity and Stability of Guinea Pig Complement. J. Immunol. 1961, 87 (6), 688–695. https://doi.org/10.4049/jimmunol.87.6.688]. In addition, zinc ions increase osmotic stability of the RBC membrane [Kabat, I.A.; Niedworok, J.; BÅ‚aszczyk, J. Der Einfluss von Zinkionen auf ausgewählte osmotische Parameter menschlicher Erythrozyten in vitro [Effect of zinc ions on the selected osmotic characteristics of human erythrocytes in vitro (author's transl)]. Zentralbl. Bakteriol. Orig. B. 1978, 166 (4-5),375-80. PMID: 654691]. It has also been demonstrated and cell adhesion [Kabat, I.A.; Niedworok, J.; BÅ‚aszczyk, J. Der Einfluss von Zinkionen auf ausgewählte osmotische Parameter menschlicher Erythrozyten in vitro [Effect of zinc ions on the selected osmotic characteristics of human erythrocytes in vitro (author's transl)]. Zentralbl. Bakteriol. Orig. B. 1978, 166 (4-5),375-80. PMID: 654691]. However, increasing the concentration of Zn2+ ions beyond the permissible limit, on the contrary, reduces these indicators. Therefore, a negative effect of ECAP-treated alloy on cells is more likely after contact with the alloy in the initial state, where degradation is significantly faster. It can be expected that the high degradation rate may lead to a decrease in biocompatibility indicators with increase of incubation time»
Regarding the in vivo study, the site of under the skin is a wet environment but supposingly poor blood flow as compared with bone and intramedullary parts. Because the degradation/corrosion rate of biodegradable alloy is much influenced by the implant site, the author should discuss this point of view supposing specific application of the proposed materials.
Answer: We appreciate the reviewer for the valuable comment. Indeed, bone implantation is more representative of the real conditions of the final implant use. However, the main task of the conducted in this work in vivo studies was to assess the overall toxicity of the material and its impact on internal organs and the tissues surrounding the implant. Currently, we are conducting the studies to assess the behavior of implants implanted in the femur bone of mice. We hope that the obtained data will be published as soon as possible.
